# Real-Life Inter-Rater Variability of the PRAETORIAN Score Values

**DOI:** 10.3390/ijerph19159700

**Published:** 2022-08-06

**Authors:** Szymon Budrejko, Maciej Kempa, Wojciech Krupa, Tomasz Królak, Tomasz Fabiszak, Grzegorz Raczak

**Affiliations:** 1Department of Cardiology and Electrotherapy, Medical University of Gdansk, 80-214 Gdansk, Poland; 2Department of Cardiology and Internal Diseases, Collegium Medicum, Nicolaus Copernicus University, 87-100 Bydgoszcz, Poland

**Keywords:** PRAETORIAN score, implantable cardioverter-defibrillator, subcutaneous implantable cardioverter-defibrillator, defibrillation testing, sudden cardiac death

## Abstract

(1) Background: The PRAETORIAN score is a tool developed for postoperative evaluation of the position of subcutaneous implantable cardioverter-defibrillator systems. The aim of our study was to evaluate the real-life inter-rater variability of the PRAETORIAN score, based on chest radiographs of S-ICD patients reviewed by independent clinical raters. (2) Methods: Postoperative chest X-rays of patients that underwent S-ICD implantation were evaluated by five clinical raters who gave values of the PRAETORIAN score. Ratings were then compared in a fully crossed manner to determine the inter-rater variability of the attributed scores. (3) Results: In total, 87 patients were included in the study. In the case of the most important final risk category of the PRAETORIAN score, the mean Light’s kappa was 0.804, the Fleiss’ kappa was 0.249, and the intraclass correlation was 0.38. The final risk category was identically determined by all five raters in 75.86% of patients, by four raters in 14.94%, and by three raters in 9.20% of patients. (4) Conclusions: The overall inter-rater variability of the PRAETORIAN score in a group of electrophysiologists experienced in S-ICD implantation, yet previously naive to the PRAETORIAN score, and self-trained in its utilization, was only modest in our study. Appropriate use of the score might require training of clinical raters.

## 1. Introduction

The implantable cardioverter-defibrillator (ICD) has been developed for secondary and primary prevention of sudden cardiac death [1]. Transvenous ICD (T-ICD) systems have their limitations, mainly associated with lead complications and infections. These limitations led to the invention of the subcutaneous ICD (S-ICD) system [2]. Initially, the S-ICD seemed to be ideal for patients not eligible for T-ICD (i.e., with no vascular access available) or who were already having complications of the T-ICD therapy (such as lead failure or infection). With its increasing availability and reimbursement as a form of therapy, the S-ICD may now be considered for all patients requiring an ICD unless they have specific contraindications (inappropriate ECG screening, need for permanent pacing due to bradycardia, cardiac resynchronization therapy, or antitachycardia pacing due to repetitive ventricular tachycardia) [1].

The S-ICD system is typically implanted in the subcutaneous tissue of the chest. The lead is tunneled along the left margin of the sternum from the region of the xiphoid process towards the manubrium of the sternum, and the device is located on the lateral surface of the chest, on or posterior to the mid-axillary line. According to current instructions from the manufacturer (Boston Scientific) available on their website, a defibrillation test (DFT) is recommended at implantation or replacement of the device and if the patient has any concomitant implants, unless the test is contraindicated in a given patient. Postoperative chest X-ray in two perpendicular projections (postero-anterior and lateral) is typically performed to determine the final location of the system, usually on the day following implantation, in standing position.

In the case of T-ICD systems, the need for DFT testing after implantation has been questioned, and, based on the results of targeted research [3], it is currently waived in many cases. DFT is still required for S-ICD systems; it is typically performed with an energy of 65J, and it may be of clinical value in selected patients [4]. Nonetheless, there are publications reporting acceptable efficacy of S-ICD systems implanted without DFT [5,6,7]. Moreover, efforts are being made to investigate the true amount of energy needed to successfully terminate ventricular fibrillation [8,9]. The test might potentially be waived completely, and the final probability of successful high-voltage therapy might be based on other, noninvasive measures [10].

According to the experimental and clinical research, it has been determined that the defibrillation threshold of the S-ICD system may be associated with the following risk factors: the final position of the system (specifically regarding the amount of fat tissue under the coil of the lead, the amount of fat tissue under the device, and placement of the S-ICD generator anteriorly on the thoracic wall) and high body mass index (BMI) [11,12,13]. Therefore, the PRAETORIAN score (PS) was developed, based on those factors, to quantify the possible risk of failed DFT [14,15].

The score was initially designed to objectively assess the final location of the system and to correlate it with DFT. It was then validated in available retrospective cohorts [14], and in separate cohorts [7,16]. Currently, the next step of research is ongoing, with the aim of skipping DFT entirely and substituting it with the score evaluation only (i.e., prospective validation of the score by randomizing to standard DFT or using only the score without conversion testing; the PRAETORIAN DFT study) [10]. Moreover, in a recent proof-of-concept study [17], the intraprocedural assessment of the score was found to be feasible.

It has been proven that the value of PS in predicting high risk of defibrillation failure is greater than the impedance measurement only, as well as the evaluation of body mass index (BMI) only. The value of PS is numerically equal to a rough estimation of the DFT in joules (based on computer modeling) [14].

As evaluation of the score incorporates several steps in which subjective decisions have to be made by the rating clinicians to assign values in categories corresponding to visual judgement, it is possible that single values and final score may be biased and may vary between raters. Therefore, the aim of our study was to evaluate the real-life inter-rater variability of the PRAETORIAN score, based on retrospectively reviewed chest radiographs of consecutive S-ICD patients treated in our departments, and ratings given by independent clinical raters.

## 2. Materials and Methods

We included consecutive patients who underwent implantation of an S-ICD system according to contemporary clinical guidelines and national reimbursement regulations in Poland, in either of our clinical centers (university hospitals in Gdansk or Bydgoszcz, Poland), from the first implantation in 2014 up to 2021. Medical records and hospital data management systems were reviewed for the values of BMI of the patients, and anonymized chest X-rays in two projections were obtained. The X-ray pictures were then initially reviewed for compliance with the basic requirements of the PRAETORIAN score algorithm (two projections available, postero-anterior and lateral, with the lead and the can visible in the picture, and located within the borderlines for the score). The anonymized pictures were then presented to five independent raters from two centers, all of whom were active electrophysiologists experienced in the implantation of all types of cardiac implantable electronic devices, including S-ICD systems. All raters were naive to the PRAETORIAN score concept and were requested to read the available publications regarding the design of the score, as well as to acquaint themselves with the available definitions and examples [8,14,16]. They were then asked to independently evaluate the X-rays for score values and to report the values for steps 1, 2, and 3 of the score, with each rater scoring all of the patients included in the study. Evaluation of the PRAETORIAN score followed the algorithm described by its authors in four steps [14]. In step 1, clinical raters determined the amount of subcoil fat by assessing the thickness of adipose tissue between the coil and the sternum or ribs on the lateral view X-ray, using the coil width as a reference (four possible values: 30, 60, 90, and 150). In step 2, they determined (again using the lateral view) whether the generator was positioned on or posterior to the midline of the chest (three possible values: x1, x2, x4). In step 3, they determined the amount of subgenerator fat by using the generator width as a reference on the postero-anterior view (two possible values: x1, x1.5). In step 4, after multiplying the value from step 1 by the values from step 2 and 3, 40 points were subtracted for patients with a BMI of 25 kg/m^2^ or less in case of a calculated score of ≥90. That gave the final PS number, and that value was then translated into categorical risk of conversion failure (below 90—low, from 90 to 150—intermediate, and 150 or more—high risk of failure).

No interobserver consultations were allowed; each clinician had to give his personal values for each of the patients. The values were then reported by the raters in Microsoft Excel spreadsheets. Next, based on the scores for each step, final values and final risk levels were computed, using the appropriate BMI values of the patients. The score matrix was constructed, including data for all patients, for whom each step of the score was evaluated by each of the observers (fully crossed design). Finally, the scores and secondary ratings were compared to determine the inter-rater variability of the score.

## 3. Statistical Analysis

Values for steps 1, 2, and 3 of the PRAETORIAN score are reported as categorical variables, according to the score design. BMI and secondary calculations are presented as continuous variables. Means with standard deviations were analyzed for continuous variables. Statistical significance was assumed at p values below 0.05. Data management and statistical analysis were performed in Microsoft Excel and R version 4.1.2 (1 November 2021, “Bird Hippie”, The R Foundation for Statistical Computing, Vienna, Austria) and R-studio software (2 September 2021 build 382). The analysis of inter-rater variability was performed according to a fully crossed design. The Fleiss’ kappa values and confidence intervals for all raters were calculated. Then, bias and prevalence adjusted kappa values for paired observers were obtained and calculated according to the Light’s method. Next, the intraclass correlation (ICC) was calculated for the subsequent steps. The inter-rater variability for each step of the score, as well as final score and final risk categories were analyzed. Finally, the rates of patients with complete and partial agreement of the scores between raters were calculated. The study design was approved by the Ethical Board at the Medical University of Gdansk, Poland.

## 4. Results

From the first implantation up to the date of our data access (1 December 2021), 110 patients underwent S-ICD system implantation in our centers (72 in Gdansk, 38 in Bydgoszcz). After data collection and validation, 87 patients were included in the study. The remaining patients were excluded due to unsuitability of X-ray pictures for our analysis (21 patients: single projections, incomplete projections partially missing the elements of the S-ICD system, picture quality insufficient to determine the score) or the lack of BMI value recorded at the time of implantation (2 patients). An exemplary set of X-ray pictures (two projections) is presented in Figure 1.

Demographic and clinical data of patients included in the study group are presented in Table 1. DFT was performed in 65 patients. In all patients, the test was successful with the first shock, performed with an energy of 65J in 61 patients and 60J in 4 patients. The test was not performed in 22 patients (25.3%), and the reasons for this decision were as follows: severe heart failure (6 patients), neurological complications of sudden cardiac arrest (3 patients), thrombus in the left atrial appendage and atrial fibrillation (3 patients), thrombus in the left ventricle (4 patients), transvenous lead extraction preceding S-ICD implantation (3 patients), noninducible ventricular arrhythmia despite repeated induction (1 patient), abdominal aortic aneurysm (1 patient), and for 1 patient the reason was not specified in the available documentation. We did not analyze more extended clinical data, as they were not relevant for the purpose of our study.

Results of the inter-rater variability (or in other words, inter-rater agreement) are presented in Table 2 and Table 3. Table 2 contains the results of the analysis of rates and percentages of agreement between raters, i.e., the numbers and percentages of patients in whom certain numbers of raters agreed in terms of the score attributed to a given patient (for categorical variables), and the distribution of values calculated for the overall score (a continuous variable calculated from the specified values of the previous three steps of the score for each patient according to each rater).

In Table 3, we present the results of the analysis of the general inter-rater variability (kappa values calculated according to Light’s method and Fleiss’ method, as well as the intraclass correlation). For the purpose of the detailed analysis of the results, Figure 2, Figure 3, Figure 4 and Figure 5 present the rates of available scores given by all five raters in general, to visualize the prevalence of specific categories of the score in the whole cohort and in the ratings of each rater.

## 5. Discussion

The PRAETORIAN score is a novel tool proposed for evaluation of the final location of S-ICD systems, intended to determine the likelihood of successful or unsuccessful defibrillation. It is hypothesized that the score may potentially replace DFT testing if its value is confirmed in the PRAETORIAN DFT trial. While awaiting the conclusive data, we attempted to assess the inter-rater variability of scoring between independent raters. If the score is to be introduced into clinical practice to guide definitive decisions regarding patient safety, it has to be reliable, repeatable, and easily applicable. In our study, we evaluated the score values given by independent, score-naive raters after self-training based on available publications, as will probably be the case for future score users.

In our fully crossed design for 87 patients and five raters, we used the most acknowledged methods for assessment of inter-rater agreement (or disagreement). As a first step, these methods included plain mathematical indexes like the percentage and rate of agreement, as well as mean and standard score deviation for continuous variables. These indexes are believed to be of limited value, and served only for to provide a general overview of the results, and not final conclusions [18].

For the rates and percentages of agreement, we used the modified indexes for five raters (Table 2). For step 1, five, four, and three identical ratings were obtained in 40.23%, 22.99%, and 27.59% of patients, respectively; for step 2—in 93.10%, 2.30%, and 3.45%; for step 3—in 77.01%, 10.34%, and 12.64% of patients, respectively. As steps 1 to 3 allow for four, three, and two distinctive values, respectively, our results show that step 2 and 3, despite having apparently higher agreement rates, may be biased by two facts. First, the lower number of categories may lead to higher chance agreement. Second, favorable ratings leading to lower scores were far more prevalent in all raters for all patients, as the incidence of clearly inappropriate placement of the system appeared to be low in our cohort. In step 1, where four categories were allowed and higher variability was observed of the possibly slanting lead course, and the anatomy of the sternum and ribs, the distribution of values was more scattered, leading to lower agreement rates. The final score was identical according to all five raters in 28 patients. In the remaining patients, the standard deviation (SD) of the scores ranged from 6.7 to 42.66, and the mean of the SD values for the patients with nonidentical scores was 17.46. The final risk category was identically determined by all five raters in 75.86% of patients, by four raters in 14.94%, and by three raters in 9.20% of patients. Based on these calculations, we might conclude that the overall agreement for the most important index, namely the risk category, was lower than for steps 2 and 3, but higher than for step 1; therefore, the score seemed to correct slightly for the inter-rater variability when reaching the final risk category.

As indexes based on percent agreement are considered insufficient for the conclusive assessment of inter-rater variability, we then proceeded to apply statistical methods specifically designed for such analyses [18,19,20]. The values of all the calculated indexes are presented in Table 3. Kappa values below 0.2 are considered to indicate slight agreement between raters, 0.21–0.4—fair, 0.41–0.6—moderate, 0.61–0.8—substantial, and 0.81–0.99—almost perfect agreement. Fleiss’ kappa is classically based on the fully crossed analysis of the whole matrix of results for all raters, while Light’s kappa uses the mean and standard deviation of paired calculations between raters.

For step 1 of the score, the Light’s kappa was 0.341, Fleiss’ kappa 0.251, and ICC 0.434, indicating fair to moderate agreement. For steps 2 and 3, we observed higher values for Light’s kappa (0.935 and 0.765) than for Fleiss’ kappa (0.283 and 0.368) and ICC (0.289 and 0.436). This is due to the fact that the prevalence of specific ratings in those two steps was high for all the raters. Despite the prevalence correction of the method, the influence of chance agreement in the paired comparisons was higher than in fully crossed analysis of more raters. Single diverging values in paired comparisons may be more expressed in fully crossed comparisons if they are distributed across different patients, rather than the same patient for all raters. This was the case in our cohort, when analyzed on a patient-by-patient basis. Step 4 had a Light’s kappa of 0.142, Fleiss’ kappa of 0.242, and ICC of 0.379, which is consistent with only slight to fair agreement.

The most important part of our analysis was the variability in the final risk category. Disagreement in the subsequent steps of any score may be negligible if it does not influence the final, clinically relevant decision. In the case of the risk category of the PRAETORIAN score, the mean kappa from paired comparisons was 0.804, but Fleiss’ kappa was 0.249 and ICC was 0.38. Again, the final risk category was identically determined by all five raters in 75.86% of patients, by four raters in 14.94%, and by three raters in 9.20% of patients; it should be noted that the distribution of categories in all raters was skewed towards low risk, hence the high percentage of agreement is not to be overestimated, as it may contain a high amount of chance agreement. Taking all the above results into account, we may conclude that paired comparisons overestimated the agreement rate for the final risk category, which turned out to be only fair according to fully crossed methods.

The above considerations also have to be commented on from the clinical perspective. According to available reports [21,22], the widely adopted intermuscular technique of implantation results in a consistently dorsal position of the device and its proximity to the costal plane (the DFT-increasing layer of subcutaneous fat tissue is left above the device by definition). Tunneling of the lead, and hence its final position in relation to the sternum and ribs, remains the most variable yet crucial part of the procedure in terms of the final position of the system and its DFT [23]. From this point of view, the position of the lead, which is step 1 of the PS evaluation, might be considered the most important part of the score, and in our analysis it seemed to be characterized by high inter-rater variability.

If the PRAETORIAN score finds its way into routine clinical practice and definitive decisions to skip DFT are to be made based on the assessment of X-ray system location only, the score has to be flawless in design and easy to use for untrained clinicians. Although the former may be true for the PS, the latter seemed not to be the case in our analysis. We speculate that this may be due to the fact that we evaluated untrained raters, based on self-education about the method. To correct for this, educational efforts will have to be made to decrease the variability of the ratings in real-life patients and to enhance the clinical safety of decisions to skip DFT based on the score result alone, possibly including score validation by multiple raters. Automated methods based on artificial intelligence to evaluate the location of the S-ICD system in an unbiased and truly objective way could be of use, and might become a future direction of research.

## 6. Limitations of the Study

We did not correlate our estimates of final risk categories with the results of the DFT. First, this was not the purpose of our study. Second, we lacked negative DFT results for comparison, as in our cohorts, all defibrillation tests that were performed ended with successful termination of ventricular fibrillation.

## 7. Conclusions

The overall inter-rater variability of the PRAETORIAN score in a group of electrophysiologists experienced in S-ICD implantation, yet previously naive to the PRAETORIAN score, and self-trained in its utilization, was only modest in our study. Appropriate clinical use of the score might require training of potential raters to reduce the inter-rater variability and to maximize the clinical reliability of the score.

## Figures and Tables

**Figure 1 ijerph-19-09700-f001:**
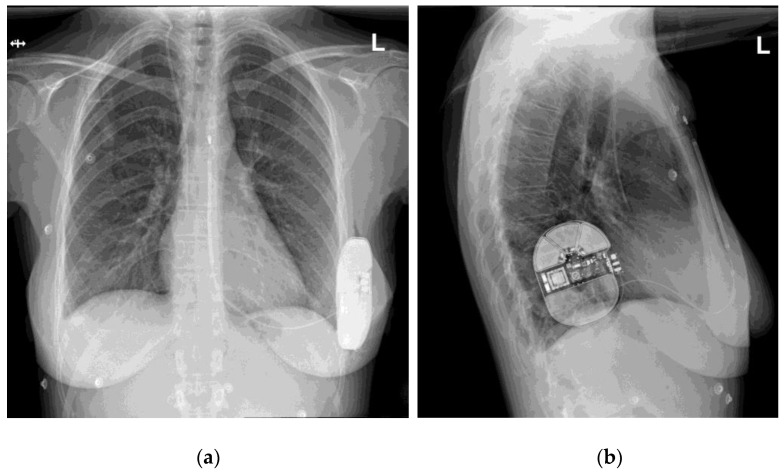
An exemplary set of X-ray pictures of the S-ICD system implanted in one of our patients, two projections: (**a**) postero-anterior view; (**b**) lateral view.

**Figure 2 ijerph-19-09700-f002:**
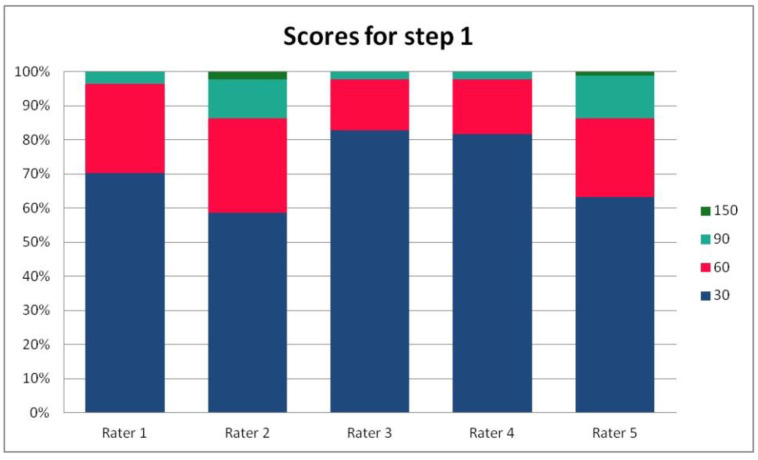
The percentage of ratings in the available categories given by individual raters for step 1 of the PRAETORIAN score.

**Figure 3 ijerph-19-09700-f003:**
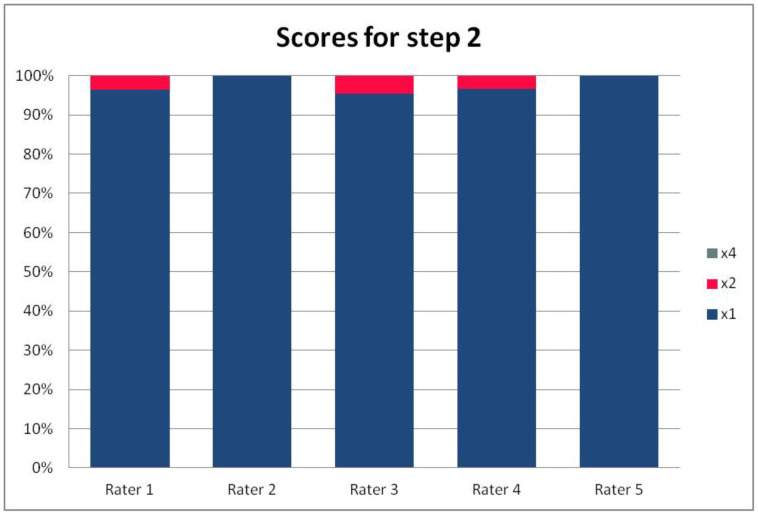
The percentage of ratings in the available categories given by individual raters for step 2 of the PRAETORIAN score.

**Figure 4 ijerph-19-09700-f004:**
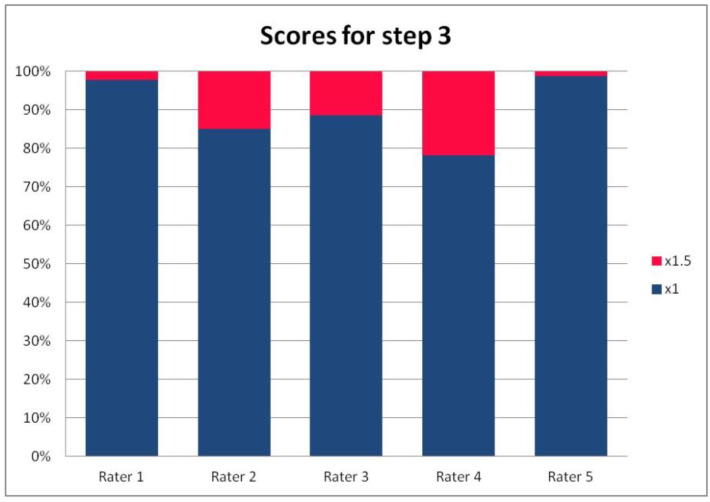
The percentage of ratings in the available categories given by individual raters for step 3 of the PRAETORIAN score.

**Figure 5 ijerph-19-09700-f005:**
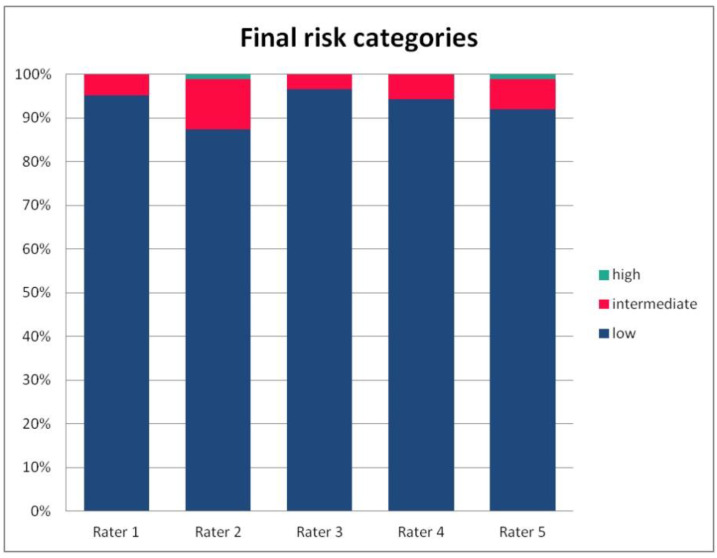
The percentage of risk categories of the PRAETORIAN score, calculated for the patients on the basis of ratings given by individual raters.

**Table 1 ijerph-19-09700-t001:** Demographic and clinical data of patients included in the study group.

sex—male	64 (73.6%)
BMI [kg/m^2^]	17.6–37.2 (26.4 ± 4.7)
age [years]	13–82 (46.2 ± 15.5)
LVEF [%]	10–75 (41.2 ± 17.2)
Prevention:	
primary	38 (44%)
secondary	49 (56%)
Etiology:	
ICM	29 (33%)
NICM	24 (28%)
IVF	20 (23%)
HCM	5 (6%)
LQTS	5 (6%)
myocarditis	2 (2%)
LVNC	1 (1%)
ARVC	1 (1%)

LVEF—left ventricle ejection fraction. ICM—ischemic cardiomyopathy. NICM—nonischemic cardiomyopathy. IVF—idiopathic ventricular fibrillation. HCM—hypertrophic cardiomyopathy. LQTS—long QT syndrome. LVNC—left ventricle non-compaction. ARVC—arrhythmogenic right ventricular cardiomyopthy.

**Table 2 ijerph-19-09700-t002:** Results of the basic analysis of inter-rater variability of the PRAETORIAN score.

Variable	Number of Raters in Agreement [N]	Number of Cases in Which [N] Raters Agreed	Percentage of Cases in Which [N] Raters Agreed
Step 1	5	35	40.23%
	4	20	22.99%
	3	24	27.58%
	2	8	9.20%
Step 2	5	81	93.10%
	4	2	2.30%
	3	3	3.45%
	2	1	1.15%
Step 3	5	67	77.01%
	4	9	10.35%
	3	11	12.64%
	2	0	0%
Step 4—overall score	in 28/87 patients—complete agreement (standard deviation of ratings in a given patient = 0); in the remaining 59 patients—SD of ratings in a given patient = 6.7–42.66 (mean 17.36).		
Risk category	5	66	75.86%
	4	13	14.94%
	3	8	9.20%
	2	0	0%

Step 1–4—subsequent steps of the PRAETORIAN score evaluation. N—number. Step 4, overall score, is a continuous variable, and therefore is presented in a separate data format.

**Table 3 ijerph-19-09700-t003:** Demographic results of the analysis of the total inter-rater variability for each step of the PRAETORIAN score evaluation.

	Light’s Kappa [mean]	Light’s Kappa [SD]	Fleiss’ Kappa	Fleiss’ Kappa 0.95 CI	ICC	ICC 0.95 CI
Step 1	0.341	0.159	0.251	0.197–0.305	0.434	0.333–0.542
Step 2	0.935	0.030	0.283	0.216–0.35	0.289	0.195–0.398
Step 3	0.765	0.113	0.368	0.301–0.434	0.436	0.331–0.545
Step 4—overall score	0.142	0.157	0.242	0.198–0.285	0.379	0.278–0.49
Risk category	0.804	0.081	0.249	0.185–0.314	0.38	0.279–0.491

Step 1–4—subsequent steps of the PRAETORIAN score evaluation. Light’s kappa—results of the kappa calculation according to the Light’s method (mean and standard deviation of paired calculations between raters). SD—standard deviation. Fleiss’ kappa—results of the Fleiss’ kappa calculation for the fully crossed analysis of five raters. ICC—intraclass coefficient calculation for the fully crossed analysis of five raters. 0.95 CI—0.95 confidence interval (lower and upper border).

## Data Availability

Anonymized data is available on demand from the corresponding author.

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
