# Peer review of "Real-Life Inter-Rater Variability of the PRAETORIAN Score Values"

_ijerph, 2022, doi:10.3390/ijerph19159700_

Round 1

Reviewer 1 Report

Budrejko et al. evaluated the real-life inter-rater variability of the PRAETORIAN score based on chest radiographs of S-ICD patients. Ther Authors report that overall inter-rater variability of the PRAETORIAN score in a group of electrophysiologists naive to the PRAETORIAN score is relatively high. 

The study is of interest but can be improved.

1.    The introduction is too long.

2.    There is no need to describe the PS in the introduction. Move this to the methods section and describe it more in depth.

3.    Ligth’s kappa values for overall score is 0.142, indicating only slight agreement. The substantial agreement (k= 0.80) for overall risk estimates is probably due to the wide limits of score categories (<90; between 90 and 150; >150) and low chances of falling into high risk categories. Fleiss kappa values also indicate slight to moderate agreement for all evaluations. Please discuss.

4.    The Authors state that “the most important part of our analysis is the variability in the final risk category. Disagreement in the subsequent steps of any score may be negligible, if it does not influence the final and clinically relevant decision”. This reviewer respectfully disagrees with this comment. Step 1 (coil depth) is the most variable and critical achievement of SICD implantation and is influenced by the implantation technique (2 or 3-incision) (J Cardiovasc Electrophysiol. 2019 Jun;30(6):854-864). SICD pulse generator proximity to the costal plane and its posterior rather than anterior position is also influenced by the implantation technique (subcutaneous vs intermuscular). However, the IM technique is nowadays largely adopted (Int. J Cardiol. 2018 Dec 1;272:162-167) and almost invariably results in appropriate can position. As implantation technique influences defibrillation efficacy (J Cardiovasc Electrophysiol. 2021 Jun;32(6):1695-1703), the Authors are encouraged to discuss their findings in the light of current literature and should discuss more in depth the implications of poor inter-observer agreement regarding coil position.

In general, my point of view is that inter-observer variability is only modest among physicians naif to PS calculation. In this view, a reasonable conclusion is that reliable PS calculation requires trained raters.

Author Response

Reviewer’s comments:

  1. The introduction is too long.

Authors’ reply: we have shortended the introduction, as suggested

  1. There is no need to describe the PS in the introduction. Move this to the methods section and describe it more in depth.

Authors’ reply: we have modified the introduction and methods, as suggested

  1. Ligth’s kappa values for overall score is 0.142, indicating only slight agreement. The substantial agreement (k= 0.80) for overall risk estimates is probably due to the wide limits of score categories (<90; between 90 and 150; >150) and low chances of falling into high risk categories. Fleiss kappa values also indicate slight to moderate agreement for all evaluations. Please discuss.

Authors’ reply: We agree with that remark. Low agreement for overall score results from the number of possible results and their intra-rater range, and that further translates into inter-rater variability. The relatively lower variance (i.e. higher agreement) for the risk category results from the low number of categories, and low overall incidence of high risk (to demonstrate that, we reported individual ratings for separate clinicians), as stated by the reviewer. Moreover, Fleiss Kappa being a fully-crossed calculation for 5 raters results in lower values than average Light’s kappa based on paired comparisons, which seems logical to us. The true question in our opinion is whether we consider the overall agreement as moderate or high, as the numerical values of agreement are confusing (low, but higher for the final risk category). Therefore we concluded, that in our analysis the PS to some extent corrected for the variability towards the final risk category, but of course we agree with the above comment. The final conclusion was modified, to reflect that issue.

  1. The Authors state that “the most important part of our analysis is the variability in the final risk category. Disagreement in the subsequent steps of any score may be negligible, if it does not influence the final and clinically relevant decision”. This reviewer respectfully disagrees with this comment. Step 1 (coil depth) is the most variable and critical achievement of SICD implantation and is influenced by the implantation technique (2 or 3-incision) (J Cardiovasc Electrophysiol. 2019 Jun;30(6):854-864). SICD pulse generator proximity to the costal plane and its posterior rather than anterior position is also influenced by the implantation technique (subcutaneous vs intermuscular). However, the IM technique is nowadays largely adopted (Int. J Cardiol. 2018 Dec 1;272:162-167) and almost invariably results in appropriate can position. As implantation technique influences defibrillation efficacy (J Cardiovasc Electrophysiol. 2021 Jun;32(6):1695-1703), the Authors are encouraged to discuss their findings in the light of current literature and should discuss more in depth the implications of poor inter-observer agreement regarding coil position.

Authors’ reply: We would like to thank for that remark, which is exactly on point, and we have not addressed that in our discourse. In our discussion we intended to underline, that despite higher variability of the steps, the final agreement was slightly higher, making the score a bit resistant to subjective judgement. But we understand, that the issue is disputable and we follow the reviewers comment. We addressed the above comments in the discussion and added relevant citations.

In general, my point of view is that inter-observer variability is only modest among physicians naif to PS calculation. In this view, a reasonable conclusion is that reliable PS calculation requires trained raters.

Authors’ reply: We modified the conclusion (as stated above, we were hesitant ourselves from the beginning on how to summarize the variability in one word, „modest” seems to be just right) and underlined the last comment of the reviewer in the text. We discussed the need for training in the last paragraph of the discussion, but as suggested by the reviewer, we emphasized it again in the conclusion.

Reviewer 2 Report

Please see Reviewers Comments file attached.

Author Response

Reviewers comment: ... This paper could however benefit from some statements (in the conclusions) about the impact of this ‘negative’ finding (i.e. relatively high inter-rater variability of the PRAETORIAN score)...

Authors’ reply: We would like to thank the reviewer for reading our manuscript and all the comments. We tried to emphasize the above issue in the last paragraph of the discussion, but in response to that comment - we added also an additional statement in conclusions.

Specific Minor Comments

Line 55-58: “According to current instructions……..” this sentence is unclear, rewriting is recommended.

Authors’ reply: We rephrased the sentence, as suggested.

Line 66-70: “Nonetheless, there are publications…..” this sentence runs over 4 lines and becomes confusing to the reader. Re writing as two sentences is recommended.

Authors’ reply: We rephrased the sentence, as suggested.

Line 76: Recommend to replace ‘can’ with ‘devise-can’ for clarity.

Authors’ reply: We corrected the phrase, as suggested by the reviewer, although to our best knowledge „device can” without a hyphen dominates over „device-can” in the literature.

Line 54: Also, for consistency replace ‘devise can’ with ‘devise-can.’

Authors’ reply: We corrected the phrase, as suggested.

Line 91: “Calculation of the score consists of four steps.” It is recommended to insert a reference to support these calculations. If it is Ref 14 as stated at the paragraph end, also add it in line 91.

Authors’ reply: The whole paragraph was rephrased and moved to methods according to the suggestion of another reviewer, but we added the reference again, as suggested, into the new location of the phrase.

Line 72-75: It is recommended to insert the word ‘patient’ after the numbers e.g. “cardiac arrest (3),” should be “cardiac arrest (3 patients)”……

Authors’ reply: We corrected all those phrases, as suggested.

Line 191: Table 2: It is suggested to use percentages (as well as numbers) in columns to provide easier comparisons. Otherwise please indicate why not.

Authors’ reply: We are not certain about the meaning of that comment. Percentages are reported in the third column, along the numbers of patients in whom N raters agreed, which are reported in column 2. So the first column states the number of raters [N] in complete agreement, the second column – in how many cases those raters fully agreed, and the third column – the percentage of all cases. To our understanding the requested data is already there.

The Discussion is mainly about the Findings from the Results, which is acceptable. However, this section could benefit from some discussion about how these findings relate to other studies, and to a clinical setting. It is noted that there are limited references cited in the Discussion section so addressing the above will support your research findings.

Authors’ reply: There are no other studies that can be related to. The score is a relatively new tool, and to our best knowledge its inter-rater variability has not yet been reported. Nonetheless, we added a paragraph regarding the clinical perspective to the discussion, as well as some additional references.

Line 312: In the Conclusions it may be beneficial to add a statement about the implications/meaning of the PRAETORIAN score being relatively high in this study. For example, what does this mean in a clinical setting? This will also need to be reflected in the main Abstract.

Authors’ reply: We supplemented the conclusions, as suggested, and they were also changed according to the remark of another reviewer. We also believe that the appropriate clinical use of the score might require training of potential clinical raters. We added that information to the abstract, as suggested.

Round 2

Reviewer 1 Report

The authors have sufficiently responded my comments. No further comments.